# Trends in the Incidence of Acute Hepatitis B in the Polish Population and Their Determinants

**DOI:** 10.3390/medicina57080738

**Published:** 2021-07-22

**Authors:** Barbara Stawinska-Witoszynska, Jan Klos, Waclaw Moryson, Barbara Wieckowska

**Affiliations:** 1Department of Epidemiology and Hygiene, Chair of Social Medicine, Poznan University of Medical Sciences, Rokietnicka 4, 60-806 Poznan, Poland; bwitoszynska@ump.edu.pl; 2Department of Public Health, Chair of Social Medicine, Poznan University of Medical Sciences, Rokietnicka 4, 60-806 Poznan, Poland; klos@ump.edu.pl; 3Department of Computer Science and Statistics, Poznan University of Medical Sciences, Rokietnicka 4, 60-806 Poznan, Poland; basia@ump.edu.pl

**Keywords:** hepatitis, incidence, epidemiology

## Abstract

*Introduction:* The World Health Assembly adopted the Global Health Strategy and aims to reduce the incidence of Hepatitis from up to 10 million cases per year to 0.9 million cases and to reduce deaths from 1.4 million to 0.5 million per year by 2030. However, given the prevalence of chronic Hepatitis B in many countries and the incidence of new cases of acute Hepatitis B, the task is not easy. This study investigates the trends and determinants of the incidence of acute Hepatitis B in Poland in 2005–2019. *Materials and Methods*: Data on the incidence of acute hepatitis B (AHBV) were obtained from the National Institute of Public Health. A case definition for AHBV was consistent with the EU definition. The incidence trends were determined by considering the sex, age and place of residence. Due to the exponential dependence model, the computations were based on the logarithm of the incidence rate. This allowed for the transformation to linear form and analysis could be conducted using linear models. Pearson’s correlation was used to determine the linear trend of incidence in general and according to sex and place of residence. The values of incidence rates (independent proportions test) and the coefficients illustrating the trends under study were also compared among males and females as well as urban and rural residents. *Results*: The incidence of AHBV in the Polish population decreased with similar slopes in both sexes. The newly reported cases of AHBV were more frequent in the male population. The incidence of acute Hepatitis B in the urban population was significantly higher than in the rural population. The significant decreasing trends in incidence were observed in all age ranges, with the exception of two age ranges 0–4 and 10–14, where the total incidence during the whole study period was negligible. *Conclusion:* Despite the significant decrease in the incidence of AHBV in Poland and its position among the European countries with the lowest hepatitis B (HBV) incidence, the alarmingly high proportion of iatrogenic infections requires further improvement in the sanitary condition of health care facilities. It is also necessary to decrease the number of unvaccinated individuals.

## 1. Introduction

Over the past 18 months, COVID-19 has dominated other infectious diseases in terms of scope, incidence and mortality rates. Some of them, however, such as the Hepatitis B virus, remain a major global health problem [1,2,3,4,5]. The level of total Hepatitis B incidence in Poland is currently influenced by chronic hepatitis [6,7]. A steady upward trend in the number of reported new chronic Hepatitis has been observed in many European Union/European Economic Area (EU/EEA) countries since 2008 [8,9].

Despite the introduction of mass vaccination and other preventive measures, the problem of acute Hepatitis B (AHBV) remains an ongoing and unresolved issue due to the lack of complete immunisation of children (85% of infants vaccinated with three doses worldwide) and new infections in unvaccinated or under-vaccinated groups [10]. In 2016, the World Health Assembly adopted the Global Health Sector Strategy (GHSS) to eliminate viral Hepatitis by 2030. The World Health Organization’s GHSS aims to reduce the incidence of Hepatitis from 6–10 million cases per year to 0.9 million cases and to reduce hepatitis B (HBV)-related deaths from 1.4 million to 0.5 million per year by 2030 [11,12]. Reaching this aim, however, is not going to be easy.

The aim of this study was to determine the trends in the incidence of acute Hepatitis B in Poland in 2005–2019 and their determinants.

## 2. Materials and Methods

Hepatitis B is a notifiable disease controlled by TESSy—the European epidemiological surveillance system. The author of the study used National Institute of Public Health-National Institute of Hygiene (NIPH- PZH) data on the incidence of acute Hepatitis B quoted in the “Infectious Diseases and Poisoning in Poland” bulletins that are published annually [11]. The information concerning newly registered cases was reported to the Epidemiology Department of NIPH- PZH by voivodeship epidemiological surveillance offices.

In Poland, a definition of an acute form of Hepatitis B was adopted in 2005 for the purpose of epidemiological surveillance, allowing for separate registration of its acute and chronic forms. The compliance of this definition with the case definition introduced by the EU in 2002 allowed comparing data from Poland with those from other countries [13,14,15]. In 2009, Poland introduced its case definition for acute Hepatitis B that is consistent with the 2008 EU case definition [13,15,16]. In 2014, a definition corresponding to the new European definition of 2012 was adopted, which considered laboratory criteria for the acute forms of the disease and criteria for the chronic or undefined forms [13,16].

From 2005 to 2019, the incidence trends for acute Hepatitis B were determined by sex, age and place of residence of those infected. The number of cases, the population by sex and age and the incidence rates for these variables in each year are not included in this study due to the large volume of the material.

Due to the exponential dependency model, the computations were based on the logarithm of the incidence rate. This allowed the transformation to a linear form and the implementation of the analysis with linear models.

Pearson’s correlation coefficient was used to determine the linear trend in the incidence of acute Hepatitis B in the Polish population with respect to the patients’ sex and place of residence. The values of incidence rates (independent proportions test) and coefficients illustrating the trends under examination were also compared among males vs. females and urban vs. rural residents.

A significance level of *p* < 0.05 was arbitrarily adopted as the criterion of statistical significance. For multiple comparisons, it was modified according to the Bonferroni correction to *p* < 0.00333.

The computations were performed using the PQStat v1.6.0 package (PQStat Software, Poznan, Poland).

## 3. Results

In Poland, the number of recorded acute Hepatitis B infections (including hepatitis B virus and hepatitis C virus mixed infections) ranged from 649 in 2005 to 45 in 2019, with crude incidence rates ranging from 1.70/100,000 to 0.12/100,000.

The incidence of acute Hepatitis B in the Polish population showed a significant decreasing trend (*p* < 0.0001), with the rate of decline being higher in the first years included in the study (Figure 1).

This phenomenon was the result of significant and very strongly decreasing trends with a similar slope in both sexes (*p* < 0.0001) (Figure 2 and Table 1). Both the strength of the exponential trend and its slope in males and females were not significantly different (Figure 2 and Table 1). The newly reported incidence of acute Hepatitis B was significantly higher in the male population (*p* < 0.0001) (Table 1).

The incidence of acute Hepatitis B among urban residents was significantly higher than among those living in rural areas (*p* < 0.0001) (Table 1). For both areas of residence, decreasing trends were maintained (*p* < 0.0001) with higher dynamics in the first years of observation (Figure 3). Both trends were very powerful and of a similar slope. The strength of the exponential trend and its slope in the urban and rural populations did not differ significantly (Figure 3 and Table 1).

Significant decreasing incidence trends were observed in all but two age groups: 0–4 and 10–14 years, in which the total incidence during the whole study period was relatively low (Table 2). In the age group of 5–9 years, the trend could not be determined with only three registered cases of acute Hepatitis B, (Table 2).

## 4. Discussion

In Poland, reporting and recording Hepatitis B infections in clinically symptomatic patients confirmed by laboratory tests began in 1979 and, until 1985, the incidence of Hepatitis B in Poland had been amongst the highest in Europe [17]. In the following years, the number of reported cases decreased, with the largest decrease (88.2%) between 1993 and 2004 when an intensive programme of prevention and control of Hepatitis B was implemented. Between 1993 and 2004, the value of the total incidence rate of Hepatitis B decreased from 34.6/100,000 to 4.1/100,000 [17]. Between 2005 and 2019, a decrease in newly diagnosed acute Hepatitis B cases was observed and their share in the total incidence of Hepatitis B decreased from 37.6% to 1.6% during this period [18].

Between 2006 and 2014 the number of reported acute HBV cases in EU/EEA countries decreased by 56.2% and the overall incidence rate dropped from 1.6 per 100,000 population to 0.7. This decline was evident in most of the countries regardless of the initial values of incidence coefficients (the highest incidence rate was observed in Bulgaria and Lithuania—6.3/100,000 and 5.1/100,000—respectively), which was the year when the universal vaccination programme was implemented (before or after 1995) or the vaccination programme. The level of reduction in HBV incidence was higher in those countries where the universal vaccination programme was implemented earlier. Four countries were not included in this analysis; Italian, Belgian and Croatian data did not allow differentiation between acute and chronic forms of HBV, whereas Liechtenstein did not provide any data. Not all countries showed a decreasing trend over this period; the level of AHBV increased in Austria, Iceland andSpain where this phenomenon was explained by an increase in AHBV among immigrants and in Portugal where the increase was observed in drug addicts and prisoners. Both these countries implemented a universal vaccination programme after 1995, although they did not have a supplementary vaccination programme in place for risk groups [19].

In Italy, which was not included in the above analysis, following the introduction of a compulsory universal vaccination programme in 1991 for newborns and adolescents aged 12, there was a marked decrease in the incidence of acute Hepatitis B in the following years [20,21]. Acute HBV incidence rate value fell from 12/100,000 to 0.4/100,000 in 2018 [21]. The decline in prevalence and incidence levels of acute HBV was particularly evident among infants, adolescents and young adults up to 24 years of age [20,21].Since the HBV vaccination was initiated 30 years ago, it is believed that all vaccinated individuals under the age of 40 are protected against both forms of HBV [21].

According to the European Centre for Disease Prevention and Control (ECDC) 2020 report, the overall incidence trend for acute HBV in the EU/EEA countries showed a steady decline between 2009 and 2018. In 2018, this form of Hepatitis represented only 10% of HBV cases and chronic Hepatitis B represented −51%. Twenty-seven countries provided information on AHBV and, considering the 20 countries that systematically reported data from 2009 to 2018, a decrease in the average incidence rate was obtained from 1.0/100,000 population in 2009 to 0.6 in 2018 [20]. In 2018, Poland, with an incidence rate of 0.1/100,000, as well as Finland and France were among the countries with the lowest AHBV incidence [22].

The positive trends of declining incidence of acute Hepatitis B observed in European countries were also reported in many other regions and countries globally, e.g., in South Korea, New Zealand (where only acute HBV is reported) and in Canada [23,24,25].

In the USA, introducing Hepatitis B vaccination in 1991 resulted in an initial decrease in its incidence. In 2010–2019, the nationwide number of registered new cases of acute Hepatitis B remained stable, while a slight increase in incidence in 2017 in some states was explained by the opioid crisis and improved epidemiological surveillance [26,27].

During the study period, in Poland, the incidence of acute Hepatitis B in men reached a higher level than in women and the rates were 1.4–3 times higher [18], which is consistent with reports from other countries, although not all reports include figures for this ratio. According to data published by ECDC, the overall male-to-female ratio in EU/EEA countries in 2018 was 1.5:1 [22].

Apart from the patients’ sex, a factor strongly affecting the prevalence of acute Hepatitis B in Poland was the place of residence, with significantly higher incidence rates among urban dwellers compared to those living in rural areas. Unfortunately, other countries do not publish such data. The only exception is Norway, where the studies on the prevalence of HBV in the years 1992–2009 pointed to the fact that acute Hepatitis B was also diagnosed more frequently among city inhabitants [28]. In Poland, between 2005 and 2019 and in all age groups including those between 15 and 75+ years of age, the trends in the incidence of acute Hepatitis B showed a significant decrease, which indicates that the improvement in the epidemiological situation was not only the result of a decrease in incidence in groups covered by the mandatory universal vaccination programme. The lack of a significant decrease in incidence in the 0–4 and 10–14 age groups or even the impossibility to determine it among the 5 year olds to 9 year olds is associated with the small sample size.

In 2018, the highest incidence of acute Hepatitis B in Poland was recorded in the age group of 45–49 (0.24 per 100,000) and among those aged 35–39 (0.22 per 100,000), while in the previous year, the highest incidence was found in the age range of 60–64 [7]. In 2018, 40% of all acute Hepatitis B cases in EU/EEA countries were reported in the younger population between 25–34 years of age [22]. This was probably due to differences in the main routes of transmission of acute HBV in Poland and EU/EEA countries in general, with heterosexual contacts (26%), iatrogenic infections (19%) and men who have sex with men (MSM) (14%) being the most common routes in 2018 [22]. In the same year and in Poland, among patients with a known routes of transmission, iatrogenic infections predominated 60.6% (20/33) followed by sexual contact (21%), including three infections in the MSM group [7]. In addition to Poland, high levels of acute HBV incidence associated with health care facilities were also found in Italy and Romania [22].

In the countries where HBV vaccination was introduced sooner, long-term declining trends in the incidence of Hepatitis B in children, adolescents and young adults were observed sooner than in the other countries [5,20,21,29,30,31].

In 2019, 85% of infants worldwide received three doses of the Hepatitis B vaccine [32]. In the same year, universal infant vaccination was already being implemented in 49 of the 53 European WHO member states [33]. The level of administration of the birth dose of the vaccine remained uneven. In the WHO African region, it was only 6% and, when administered in time, it effectively reduced the risk of vertical infections [32].

The long-term decreasing trend in the incidence of AHBV and the place of Poland in the group of European countries with the lowest incidence rate of acute Hepatitis B is encouraging. This is the result of modifications and the expansion of protective vaccination coverage that was initiated in Poland in 1989 when only the newborns and infants born to HBV positive mothers were covered by four doses of vaccination against Hepatitis B [34]. A universal Hepatitis B vaccination programme for all newborns and infants began in 1994–1996, whereas a booster vaccination programme for junior high school students aged 14, covering the children who had not been vaccinated in their first year of life, was implemented in 2000–2011 [34,35].

Apart from universal infant vaccinations, in Poland, mandatory vaccinations were performed for many years for those who were particularly exposed to infection. This included all unvaccinated students of medical schools and other schools providing education in medical faculties; all students of medical universities or other universities providing education in medical faculties; all medical professionals exposed to infection; and all those who were particularly exposed to infection as a result of contact with Hepatitis B positive patients and Hepatitis C positive patients [35,36]. In 2019, that also included those in advanced stage kidney disease with glomerular filtration rates below 30 mL/min as well as dialysis patients [36]. Unfortunately, the percentage of newborns who are not inoculated with the first dose of HBV and tuberculosis vaccines has increased alarmingly in recent years (6.7% in 2016, 10.1% in 2017) [37].

Most EU/EEA countries and the UK recommend universal childhood HBV vaccination. However, Denmark, Finland and Iceland do not have national universal vaccination policies in place, while universal vaccination is regulated regionally in Sweden [38].

Only the wealthiest countries with low endemicity (HBsAg (surface antigen of the hepatitis B virus) < 1%) have chosen to opt for a selective vaccination programme that applies to people at a high risk of HBV infection, yet the definition of high risk varies amongst countries. The programme includes but is not limited to intravenous drug users and their community, MSM, the individuals exposed to close family contacts with an infected person, health care workers, police officers, firefighters, newborns to mothers with a chronic HBV infection or those with acute Hepatitis B during pregnancy [39].

In Poland, in addition to HBV screening for pregnant women in the third trimester of pregnancy (since 2011), children of HBV-positive mothers, patients requiring chronic haemodialysis treatment and blood donor candidates are screened [40]. Active and passive prophylaxis of vertical HBV infections in Poland consists of administering anti-HBV immunoglobulin and the first dose of HBV vaccine to the newborns of HBV-positive mothers within twelve hours after birth [41]. Blood donor testing is the most common HBV screening test performed in EU/EEA countries [42]. According to ECDC data, in 2016–2017 in the EU/EEA countries, 15 countries had a national policy that included screening for HIV-infected people. Drug users were screened in 13 countries, 12 countries screened their health workers, 11 countries screened prisoners, 6 countries screened pregnant women and 7 countries screened immigrants [43].

Although, since the 1990s, Poland has achieved significant results in the fight against HBV, there is still much to perform in order to reach the WHO targets for the elimination of viral Hepatitis [11].

## 5. Conclusions

Despite the significant downward trend in the incidence of acute Hepatitis B in Poland and its membership in the group of European countries with its lowest incidence, the alarmingly high proportion of infections associated with health care activities calls for further improvement in the sanitary condition of health care facilities. The number of people who have been immunised needs to be increased and the high level of infant immunisation needs to be maintained.

## Figures and Tables

**Figure 1 medicina-57-00738-f001:**
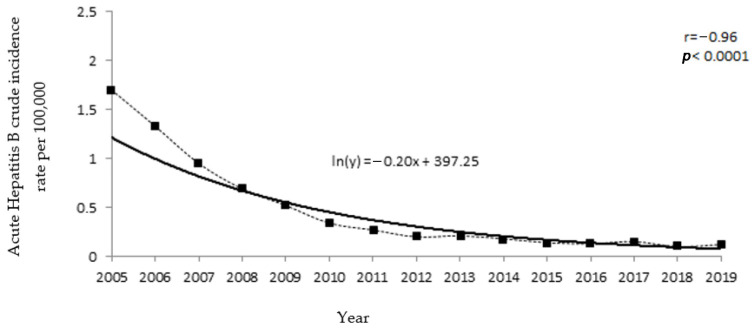
Polish population’s acute Hepatitis B incidence trends in 2005–2019.

**Figure 2 medicina-57-00738-f002:**
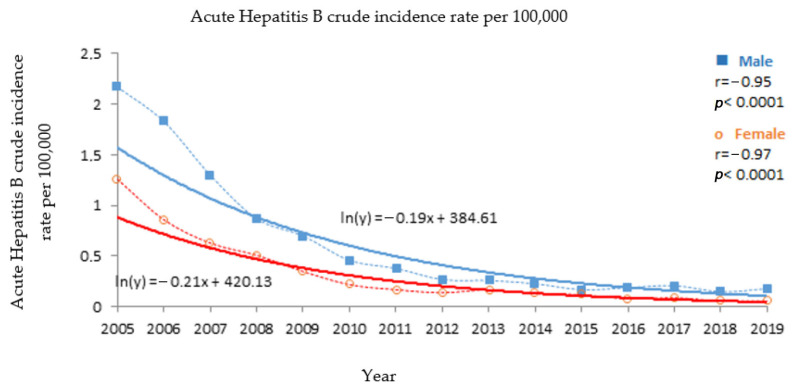
Acute Hepatitis B incidence trends for males and females in Poland, 2005–2019.

**Figure 3 medicina-57-00738-f003:**
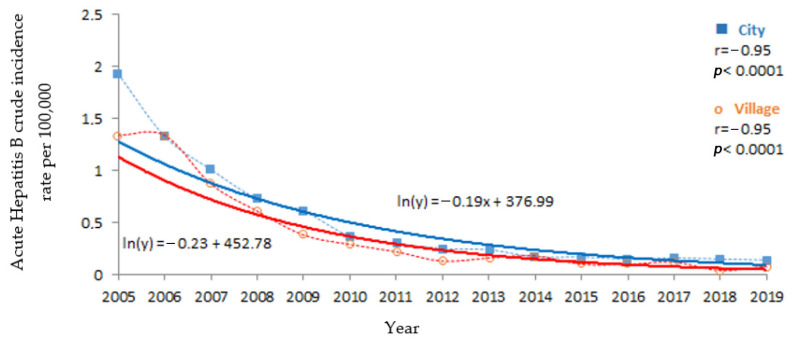
Acute Hepatitis B incidence trends amongst urban and rural residents in Poland, 2005–2019.

**Table 1 medicina-57-00738-t001:** Comparison of the direction and strength of trends in acute Hepatitis B incidence in Poland according to sex and place of residence, 2005–2019.

	Crude Rates Comparison		Comparison of the Exponential Time Trends			
	Crude Rate	*p*-Value	Slope Coefficient	*p*-Value	Pearson’s Correlation Coefficient	*p*-Value
Female	0.32	<0.0001	−0.21	0.4357	−0.97	0.4537
Male	0.62		−0.19		−0.95	
City	0.52	<0.001	−0.19	0.1574	−0.95	0.9764
Village	0.39		−0.23		−0.95	

**Table 2 medicina-57-00738-t002:** Linear trend analysis of the incidence of acute Hepatitis B in different age groups in the Polish population in 2005–2019.

Age Groups	*n*	Pearson’s Correlation Coefficient	Slope Coefficient	*p*-Value Benajamini Corected
0–4	6	−0.50	−6.68	0.3912
5–9	3	NA	Too few infected	
10–14	19	−0.91	−3.50	0.1010
15–19	54	−0.72	−2.59	0.0224
20–24	204	−0.92	−2.89	0.0002
25–29	298	−0.90	−2.89	0.0000
30–34	254	−0.91	−5.80	0.0000
35–39	236	−0.77	−3.48	0.0010
40–44	190	−0.92	−4.51	0.0000
45–49	217	−0.88	−5.08	0.0000
50–54	220	−0.82	−3.40	0.0003
55–59	204	−0.94	−3.65	0.0000
60–64	183	−0.88	−3.23	0.0000
65–74	331	−0.96	−4.01	0.0000
75+	268	−0.86	−3.29	0.0001

The significance level for the *p*-value in the table above was modified according to Bonferroni adjustment and equaled 0.0033.

## Data Availability

Data are available in a publicly accessible repository that does not issue DOIs. Publicly available datasets were analyzed in this study. This data can be found here: http://wwwold.pzh.gov.pl/oldpage/epimeld/index_p.html (accessed on 17 May 2021).

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
