# Peer review of "Trends in the Incidence of Acute Hepatitis B in the Polish Population and Their Determinants"

_medicina, 2021, doi:10.3390/medicina57080738_

Round 1
Reviewer 1 Report
The study analyzed the incidence of acute Hepatitis B in the Polish population. The study was well designed and the results were well interpreted. My only minor comment is to clarify what is x and y in the linear equations so that the readers can understand the figures better.
Author Response
Dear Sir/Madam
Thank you very much for your review.
I consider your remarks very valuable. I have modified the content of the manuscript and clarified what is x and y in the linear equations. I believe that thanks to the advice provided, the work becomes clearer for the readers.
Reviewer 2 Report
This manuscript analyses the trend of acute hepatitis B in Poland for the last 15 years. The results are interesting and very well discussed on the large field of European trend of acute hepatitis B.
The paper is well written, well organized, based on important and reliable data, well discussed.
Still, some improvements should be made.
All over the paper, the abbreviated words must be defined at their first use and used correctly (see AHBV, HBV, HC, ECDC, TB, even though some are well known).
In the abstract, there is a mention of iatrogenic infections in the Conclusions, but there is no data on this issue in the Results.
Material and methods: company, city, county for the producer of the statistical package should be mentioned.
In Discussions, maybe it would be better to include some data of the causes of acute hepatitis B if iatrogenic causes were mentioned. There are only references to the screening programs in people exposed to HBV infection due to health care activities.
The Conclusions are supported by data presented (only with the previous comment).
The study was based on many references, important and recent published (35 of 44 during the last 5 years).
Author Response
Dear Sir/Madam
Thank you very much for your review and valuable comments. I have modified the content of the article to implement you suggestions.
Please find my detailed replies to the individual passages of the article you have indicated. The changes were also made in the paper.
All over the paper, the abbreviated words must be defined at their first use and used correctly (see AHBV, HBV, HC, ECDC, TB, even though some are well known).
All the abbreviations have been defined at their first use.
In the abstract, there is a mention of iatrogenic infections in the Conclusions, but there is no data on this issue in the Results.
This is due to the fact that the data at our disposal did not contain information on the route of transmission of the infection. All information on transmission routes has been taken from the studies:
European Centre for Disease Prevention and Control. Hepatitis B. In: ECDC. Annual epidemiological report for 2018. Stockholm: ECDC; 2020.
Wiktor A, Stępień M. Hepatitis B in Poland in 2018. Przegl Epidemiol 2020;196–208.
Those studies are referenced in our publication
3.
Material and methods: company, city, county for the producer of the statistical package should be mentioned.
Information has been added to the material and methods.
4.
In Discussions, maybe it would be better to include some data of the causes of acute hepatitis B if iatrogenic causes were mentioned. There are only references to the screening programs in people exposed to HBV infection due to health care activities.
The data at our disposal did not contain information on the route of transmission of the infection. All information on transmission routes has been taken from studies referenced in our publication.
Information on transmission routes in the EU / EEA countries and Poland was cited by comparing the incidence levels in different age groups and the differences in this respect between Poland and other EU / EEA countries. The information described is included in the Discussion and the indicated paragraph takes the following form:
In 2018, the highest incidence of acute Hepatitis B in Poland was recorded in the age group of 45- 49 (0.24 per 100,000) and among those aged 35-39 (0.22 per 100,000), while in the previous year, the highest incidence was found in the age range of 60-64. In 2018, 40% of all acute Hepatitis B cases in EU/EEA countries were reported in the younger population, between 25-34 years of age. This was probably due to differences in the main routes of transmission of acute HBV in Poland and EU/EEA countries in general, with heterosexual contacts (26%), iatrogenic infections (19%) and Men who have sex with men (MSM) (14%) being the most common routes in 2018 [22]. In the same year in Poland, among patients with a known route of transmission, iatrogenic infections predominated 60.6% (20/33), followed by sexual contact (21%), including three infections in the MSM group. Besides Poland, high levels of acute HBV incidence associated with health care facilities were also found in Italy and Romania.